# Incidence of oncogenic HPV infection in women with and without mental illness: A population-based cohort study in Sweden

Eva Herweijer[1][☉], Kejia Hu[1][☉]*, Jiangrong Wang[2], Donghao Lu[1], Pär Sparén[3], Hans-Olov Adami[3,4], Unnur Valdimarsdóttir[1,5,6], Karin Sundström[2], Fang Fang[1]*

1 Institute of Environmental Medicine, Karolinska Institutet, Stockholm, Sweden, 2 Center for Cervical Cancer Elimination, Department of Clinical Sciences, Intervention and Technology, Karolinska Institutet, Stockholm, Sweden, 3 Department of Medical Epidemiology and Biostatistics, Karolinska Institutet, Stockholm, Sweden, 4 Clinical Effectiveness Group, Institute of Health and Society, University of Oslo, Oslo, Norway, 5 Centre of Public Health Sciences, Faculty of Medicine, University of Iceland, Reykjavik, Iceland, 6 Department of Epidemiology, Harvard T.H. Chan School of Public Health, Boston, Massachusetts, United States of America

☉ These authors contributed equally to this work.
* kejia.hu@ki.se (KH); fang.fang@ki.se (FF)

**Data Availability Statement:** The data underlying the results presented in the study are available from Statistics Sweden (www.scb.se/en. Contact details: Email scb@scb.se, Telephone +46 10 479

## Abstract

### Background

Women with mental illness experience an increased risk of cervical cancer. The excess risk is partly due to low participation in cervical screening; however, it remains unknown whether it is also attributable to an increased risk of infection with human papillomavirus (HPV). We aimed to examine whether women with mental illness had an increased infection rate of HPV compared to women without mental illness.

### Methods and findings

Using a cohort design, we analyzed all 337,116 women aged 30 to 64 and living in Stockholm, who had a negative test result of 14 high-risk HPV subtypes in HPV-based screening, during August 2014 to December 2019. We defined women as exposed to mental illness if they had a specialist diagnosis of mental disorder or had a filled prescription of psychotropic medication. We identified incident infection of any high-risk HPV during follow-up and fitted multivariable Cox models to estimate hazard ratios (HR) with 95% confidence intervals (CI) for HPV infection.

A total of 3,263 women were tested positive for high-risk HPV during follow-up (median: 2.21 years; range: 0 to 5.42 years). The absolute infection rate of HPV was higher among women with a specialist diagnosis of mental disorder (HR = 1.45; 95% CI [1.34, 1.57]; $p < 0.001$) or a filled prescription of psychotropic medication (HR = 1.67; 95% CI [1.55, 1.79]; $p < 0.001$), compared to women without such. The increment in absolute infection rate was noted for depression, anxiety, stress-related disorder, substance-related disorder, and ADHD, and for use of antidepressants, anxiolytics, sedatives, and hypnotics, and was consistent across age groups.

40 00) and The Swedish National Board of Health and Welfare (www.socialstyrelsen.se/en. Contact details: Email socialstyrelsen@socialstyrelsen.se, Telephone +46 (0)75 247 30 00). Updated contact information to access the data can be found through the websites. The raw datasets are however unavailable for sharing because of privacy policies and regulations in Sweden. Researchers may however contact the above data holders and apply for the use of such data in their research.

**Funding:** This work was supported by the Swedish Cancer Society (cancerfonden.se) [grant number 20 0846 PjF to FF and 23 2741 Pj]. The funders had no role in study design, data collection and analysis, decision to publish, or preparation of the manuscript.

**Competing interests:** KS has received research grants and consultation fee from Merck to her affiliating institution for research on HPV vaccination. All other authors have declared no conflicts of interest.

**Abbreviations:** ADHD, attention-deficit hyperactivity disorder; ATC, Anatomical Therapeutic Chemical; CI, confidence interval; CIN3+, cervical intraepithelial neoplasia grade 3 or worse; HPV, human papillomavirus; HR, hazard ratio; LISA, the Swedish national longitudinal integration database for health insurance and labor market studies; NKCx, Swedish National Cervical Screening Registry; STROBE, the Strengthening the Reporting of Observational Studies in Epidemiology; SVEVAC, the Swedish Vaccination Register; WHO, World Health Organization.

The main limitations included selection of the female population in Stockholm as they must have at least 1 negative test result of HPV, and relatively short follow-up as HPV-based screening was only introduced in 2014 in Stockholm.

## Conclusions

Mental illness is associated with an increased infection rate of high-risk HPV in women. Our findings motivate refined approaches to facilitate the WHO elimination agenda of cervical cancer among these marginalized women worldwide.

## Author summary

### Why was this study done?

- Mental illness has been associated with a higher risk of cervical cancer and precancerous lesions as well as a lower degree of participation in cervical screening.

- Little is known, however, regarding disparities in HPV infection between women with and without mental illness.

### What did the researchers do and find?

- In a cohort study, we followed all 337,116 women who were at age 30 to 64, living in Stockholm, and had a negative test result of high-risk HPV during August 2014 to December 2019, to assess the link between mental illness and risk of infection with high-risk HPV.

- The absolute infection rate of HPV was 45% higher among women with a specialist diagnosis of mental disorder and 67% higher among women with a filled prescription of psychotropic medications, compared to women without such.

### What do these findings mean?

- Mental illness is associated with an increased infection rate of oncogenic HPV in women.

- Refined approaches are needed to facilitate the elimination agenda of cervical cancer among women with mental illness.

## Introduction

Cervical cancer, primarily caused by infection with oncogenic human papillomavirus (HPV) on the cervix [1], is preventable through HPV vaccination, which prevents HPV infection, and cervical screening, which enables early detection and treatment of cervical precancerous lesions. In 2020, the World Health Organization (WHO) launched efforts to accelerate the elimination of cervical cancer as a public health issue [2]. However, women with mental illness

present a unique challenge to this agenda, as they not only face an increased risk of cervical cancer but also tend to have reduced participation in cervical screening [3–6].

This heightened risk among women with mental illness cannot be solely attributed to lower screening rates. Even among those who participate in screening, women with specialist diagnoses of mental disorders still carry a 2-fold risk of developing cervical precancerous lesions [3]. This suggests that mental illness likely influences the risk of cervical cancer also through other mechanisms apart from differential screening participation, including modulating the risk of infection with HPV, the primary cause of cervical cancer.

Several factors may contribute to the elevated risk of HPV infection among women with mental illness, encompassing abnormal levels of immune biomarkers [7], engagement in more risky sexual behaviors [8], higher smoking rate [9], limited knowledge about HPV infection [10], and a higher chance of having experienced sexual abuse [11], compared to their counterparts. Despite these concerns, population-based studies examining HPV infection disparities between women with and without mental illness remain limited.

To address this gap, we conducted a study using data from a substantial prospective cohort of women participating in HPV-based cervical screening in Stockholm, Sweden. Our hypothesis was that women with mental illness would exhibit an increased risk of HPV infection.

## Methods

### Cervical screening in Sweden

In Sweden, a national cervical screening program with cytology has been recommended for women between the age of 23 and 60 since the 1970s [12]. In 2008, the official European guidelines classified HPV-based screening as an evidence-based screening modality [13]. A randomized implementation of HPV-based screening was initiated in the greater Stockholm area in 2012 [14,15], where all women living in the area under screening ages were randomly assigned to the invitation of either primary cytology-based screening with HPV test as triage for women with low-grade cytology, or primary HPV-based screening with cytology as triage for women with positive HPV test. Greater Stockholm area has around 1 million women, representing 20% of the entire women population in Sweden. The trial initially focused on women at age 56 to 60 but later expanded into a full-scale randomized implementation that encompassed women at age 30 to 64 starting from August 2014. Women are invited by letter to their randomized screening modality when 3 years (for women at age 30 to 49) or 5 years (for women at age 50 to 64) have passed since their last negative test was taken; reminder invitations are sent every year to women who did not attend after being invited during the last year [14].

As a result of this trial, the Swedish national screening guidelines were updated in 2015 recommending HPV test as the primary screening method [14–16]. As a result, beginning in 2017, all women between the ages of 30 and 64 in the greater Stockholm area underwent primary screening using HPV-based testing. The screening test covers 14 high-risk HPV types, including HPV-16, 18, 31, 33, 35, 39, 45, 51, 52, 56, 58, 59, 66, and 68. All data on HPV-based or cytology-based cervical screening as well as diagnosis of precancerous lesions are included in the Swedish National Cervical Screening Registry (NKCx), with nationwide coverage since 1995 [17].

### Study population

Based on the randomized implementation of HPV-based screening in Stockholm, we defined our study period as August 1, 2014 to December 31, 2019. We first identified from NKCx a cohort of women who were 30 to 64 years and living in the greater Stockholm area and had negative test results for HPV during this study period. To ensure that women were followed

after the start of the full-scale screening program, women entered the cohort on the date of their first negative HPV test, 30th birthday, or August 1, 2014, whichever occurred last. We ended the follow-up at a positive test for high-risk HPV, 65th birthday, 1 screening interval (3 years at age 30 to 49 and 5 years at age 50 to 64 [16]) plus 6 months since the last negative test for high-risk HPV (i.e., proxy for loss to follow-up), emigration out of Sweden, domestic emigration out of Stockholm, death, or December 31st, 2019, whichever came first. Six months were added in the ascertainment of loss to follow-up, allowing delays in screening invitation and participation. By such a design, women who had previously been tested positive for high-risk HPV entered the cohort the day when they tested negative for high-risk HPV, whereas women with only positive tests for high-risk HPV during the study period were excluded from the study.

To ensure that there was no undetected HPV status due to unattended screening, we defined a subset cohort of women with at least 2 HPV tests within the entire study cohort, with the first being negative test and follow-up period was defined to be between the 2 tests.

## HPV infection

The primary outcome was infection with any high-risk HPV. The secondary outcomes were infection with HPV-16/18 or other high-risk HPV, as HPV-16/18 have been estimated to contribute to approximately 70% of all cervical cancer cases [18]. The Cobas 4800 HPV Test (Roche Molecular Systems, South Branchburg, New Jersey, USA) has been used for HPV-based screening in Sweden, and test results are reported as positive or negative in "HPV-16," "HPV-18," and "other high-risk HPV." This information was obtained through NKCx.

## Mental illness

We first identified specialist diagnoses of mental disorders, including psychiatric and neurodevelopmental disorders, through linking the study cohort to the Swedish Patient Register, using the eighth, ninth, and 10th Swedish revisions of the International Classification of Diseases codes (ICD) (S1 Table). The Patient Register includes only hospital-based diagnoses made by a specialist since 1964 [19]. We used both main and secondary diagnoses at a hospital visit to identify mental disorders. Psychiatric disorders (i.e., depression, anxiety, stress-related disorder, substance abuse, and psychotic disorder) and neurodevelopmental disorders (i.e., attention-deficit hyperactivity disorder (ADHD), autism spectrum disorder, and intellectual disability) were studied separately as well as collectively as any mental disorder. We then identified filled prescriptions of psychotropic medications, through linking the study cohort to the Swedish Prescribed Drug Register, using the Anatomical Therapeutic Chemical (ATC) classification codes (S2 Table). The Prescribed Drug Register includes information on all prescribed medications (through specialist or primary care) in Sweden since July 1, 2005. We included as psychotropic medications antidepressants, anxiolytics, sedatives and hypnotics, and antipsychotics. Both a specialist diagnosis of mental disorder and filled prescription of psychotropic medication were treated as time-varying variables, namely, that women were considered exposed ever since their first diagnosis of mental disorder or first dispensation of psychotropic medication.

## Covariables

Information on year and country of birth (Nordic countries or other) was collected from the Total Population Register. The highest educational attainment at cohort entry (low, medium, high, or unclassified) was ascertained from the Swedish national longitudinal integration database for health insurance and labor market studies (LISA) [20]. HPV vaccination with at least

1 dose was identified from the Swedish Vaccination Register (SVEVAC) [21] and the Swedish National Vaccination Register [22] and treated as a time-varying covariate. Last, we identified mothers of the study participants from the Swedish Multi-Generation Register and defined maternal history of cervical intraepithelial neoplasia grade 3 or worse (CIN3+), including diagnosis of intraepithelial neoplasia grade 3 and adenocarcinoma in situ according to NKCx and diagnosis of cervical cancer according to NKCx and the Swedish Cancer Register [23], as a time-varying covariate.

## Statistical analysis

To ensure generalizability of our findings, we first compared the prevalence of mental illness in the study cohort (i.e., women at screening age who were tested negative for high-risk HPV at cohort entry) to the prevalence in the alternative female population at screening age in Stockholm (i.e., entire Stockholm female population after excluding our study cohort). We calculated prevalence through dividing exposed proportion of follow-up time by the total follow-time during the study period of both the study cohort and the alternative female population in Stockholm.

To avoid any undetected HPV status during the follow-up, we, in a subset cohort, performed an analysis by treating data as interval censored, i.e., the infection could have happened any time before a positive test since the preceding negative test. In this analysis, we included only women with at least 2 tests, with the first being a negative test. We started the follow-up from the negative test and ended the follow-up with the first positive test, or the last negative test if there was no positive test. We fitted Cox models to estimate the hazard ratios (HRs) and 95% confidence intervals (CIs) and estimated the standard errors using bootstrapping.

Within the entire study cohort, participant characteristics were summarized by the distribution of person-years among women with or without mental illness, using time-varying exposure and other time-varying covariables. Treating data as right censored, we fitted multivariable Cox proportional hazards model to estimate the relative infection rate of high-risk HPV in relation to diagnosis of mental disorder or use of psychotropic medication, as HR and 95% CI. We reported p-values of the coefficient using likelihood ratio test. We also fitted separate Cox models for HPV-16/18 and other high-risk HPV infection, as positivity of different types of HPV might be reported in 1 HPV test. We employed attained age as the underlying time scale for our analysis. No major violation of the proportional hazard assumption was found after plotting the Schoenfeld residuals over time. Models were implicitly adjusted for age (as age was used as the underlying time scale) and further adjusted for country of birth, educational level, HPV vaccination status, and maternal history of CIN3+. Finally, to examine potentially differential attendance in the subsequent screening between women with or without mental illness, we defined women who ended the follow-up at 1 screening interval plus 6 months as loss to follow-up (i.e., missing a subsequent screening) and estimated HR of loss to follow-up in relation to mental illness, using Cox models with the same adjustment.

To examine potential effect modification by age, we fitted interaction models by adding an interaction term between age (30 to 39, 40 to 49, and 50 to 64 years) and mental illness. We tested for the statistical significance of the interaction by log-likelihood ratio test. To assess the role of severity of mental illness on HPV infection rate, we performed an additional analysis by analyzing separately women with a specialist diagnosis of mental disorder (regardless of medication use) and women with only filled prescription of psychotropic medications (i.e., without a specialist diagnosis). We further performed a sensitivity analysis by censoring at 1 screening interval plus 12, instead of 6, months since the last negative test for high-risk HPV, to allow more time for delay in attending the next screening.

A statistical analysis plan written before analyzing the data can be found in the Supporting information (S1 Statistical Analysis Plan). All analyses were performed as planned. Data were managed in SAS statistical software version 9.4 (SAS Institute, Cary, North Carolina), and statistical analysis was performed in Stata version 17 (StataCorp, LP, College Station, Texas) and R version 4.3.1. The study was approved by the Swedish Ethical Review Authority (Dnr 2013/244-31/4), and informed consent by the study participants was not requested for register-based studies in Sweden. This study is reported as per the Strengthening the Reporting of Observational Studies in Epidemiology (STROBE) guideline (S1 STROBE Checklist).

## Results

The study cohort included 337,116 women with a median follow-up of 2.21 years, corresponding to 779,142 person-years (Fig 1). The proportion of person-time exposed to a specialist diagnosis of mental disorder was slightly lower in the study cohort compared with the alternative general female population in Stockholm (14.20% and 16.40%). A similar pattern was found for use of psychotropic medications (42.37% and 44.41%) (Table 1). Depression, anxiety, and stress-related disorders were the most common diagnoses among all mental disorders (Table 1).

Within the study cohort, women with mental illness were younger and had lower educational level, compared to women without mental illness. Women with filled prescriptions of psychotropic medications were older, but also had lower educational level, compared to women without such use. Country of birth, HPV vaccination, and maternal history of CIN3 + were generally similar between women with or without a diagnosis of mental disorder or filled prescriptions of psychotropic medication (S3 Table).

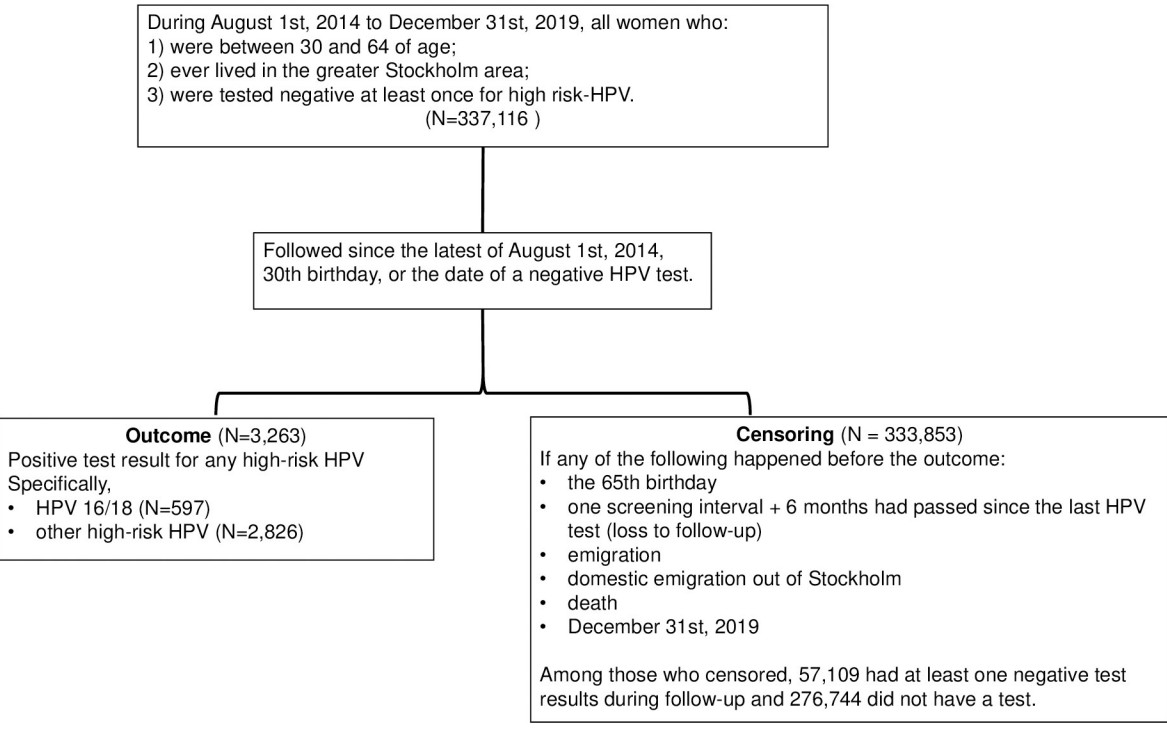

**Fig 1. Flowchart of the study design.**

Table 1. Prevalence of mental illness in the study cohort and the alternative female population of the greater Stockholm region.

| Group | Proportion of mental illness (%) | |
|---|---|---|
| | Study cohort | Alternative cohort in Greater Stockholm region |
| **Any diagnosis of mental disorder** | 14.20 | 16.40 |
| **Any diagnosis of psychiatric disorder** | 13.91 | 15.96 |
| Depression | 6.78 | 8.02 |
| Anxiety | 6.62 | 7.63 |
| Stress-related disorder | 5.74 | 6.10 |
| Substance abuse disorder[1] | 2.33 | 3.45 |
| Psychotic disorder | 0.72 | 1.11 |
| **Any diagnosis of neurodevelopmental disorder** | 1.46 | 2.07 |
| Attention-deficit hyperactivity disorder (ADHD) | 1.22 | 1.63 |
| Intellectual disability | 0.12 | 0.26 |
| Autism | 0.33 | 0.48 |
| **Any use of psychotropic medication** | 42.37 | 44.41 |
| Hypnotics and sedatives | 24.73 | 26.91 |
| Anxiolytics | 25.12 | 27.16 |
| Antidepressants | 26.99 | 29.21 |
| Antipsychotics | 3.50 | 4.81 |

[1]Substance abuse disorder includes alcohol-related, tobacco-related, and other substance abuse disorders.

Within the study cohort, 3,263 women were tested positive for high-risk HPV during follow-up. A total of 333,853 women were censored, among whom 57,109 had at least 1 negative test results during follow-up and 276,744 did not have a test (Fig 1). Among women with at least 2 HPV tests during the study period, the incidence of high-risk HPV infection was higher among women with a diagnosis of mental disorder (HR = 1.45; 95% CI [1.34, 1.57]; $p < 0.001$) or filled prescription of psychotropic medication (HR = 1.67; 95% CI [1.55, 1.79]; $p < 0.001$), compared to women without such (Table 2). The association was similar for psychiatric disorders (HR = 1.44; 95% CI [1.32, 1.57]; $p < 0.001$) and neurodevelopmental disorders (HR = 1.47; 95% CI [1.19, 1.83]; $p < 0.001$). Analyses on individual diagnoses showed an increased infection rate of high-risk HPV among women with a diagnosis of depression, anxiety, stress-related disorder, alcohol-related disorder, substance abuse (excluding tobacco or alcohol use), ADHD, or intellectual disability (Table 2). An increase in infection rate was suggested but not statistically significant for tobacco-related disorders, whereas no risk alteration was noted for psychotic disorder and autism spectrum disorder. Analyses of individual psychotropic medications showed similarly increased infection rate of high-risk HPV for sedatives and hypnotics, anxiolytics, antidepressants, and antipsychotics (Table 2). Results from unadjusted models were similar (Table 2).

Among all women within the study cohort, results from the right censored data analysis were largely similar to the ones obtained in the interval censored analysis, although the estimated HRs were slightly decreased (Fig 2 and Table 3). We also found similar results between HPV-16/18 and other high-risk HPV types and between adjusted and unadjusted models (Fig 2 and Table 3).

Women with mental illness were more likely to miss a subsequent screening (i.e., loss to follow-up) compared with women without mental illness (Table 4). The differential loss to follow-up was noted across all diagnoses of mental disorders and all kinds of psychotropic medications, although women with substance abuse (excluding alcohol and tobacco use) had

**Table 2. HRs with 95% CIs of any high-risk HPV infection in relation to diagnosis of mental disorder or use of psychotropic medication, using interval censored data.**

| Exposure | Unadjusted HRs[1] | P value for unadjusted HR | Adjusted HRs[1] | P value for adjusted HR |
|---|---|---|---|---|
| **Any diagnosis of mental disorder** | 1.48 (1.36, 1.62) | <0.001 | 1.45 (1.34, 1.57) | <0.001 |
| **Any diagnosis of psychiatric disorder** | 1.47 (1.34, 1.62) | <0.001 | 1.44 (1.32, 1.57) | <0.001 |
| Depression | 1.43 (1.24, 1.64) | <0.001 | 1.39 (1.23, 1.59) | <0.001 |
| Anxiety | 1.46 (1.29, 1.65) | <0.001 | 1.40 (1.24, 1.58) | <0.001 |
| Stress-related disorder | 1.46 (1.30, 1.65) | <0.001 | 1.47 (1.31, 1.66) | <0.001 |
| Alcohol-related disorder | 2.09 (1.69, 2.60) | <0.001 | 2.02 (1.63, 2.50) | <0.001 |
| Tobacco-related disorder | 1.47 (0.61, 3.53) | <0.001 | 1.43 (0.08, 24.17) | 0.806 |
| Substance abuse[2] | 2.05 (1.44, 2.91) | <0.001 | 1.92 (1.38, 2.68) | <0.001 |
| Psychotic disorder | 0.95 (0.61, 1.46) | 0.808 | 0.97 (0.56, 1.69) | 0.917 |
| **Any diagnosis of neurodevelopmental disorder** | 1.63 (1.31, 2.04) | <0.001 | 1.47 (1.19, 1.83) | <0.001 |
| ADHD | 1.69 (1.35, 2.12) | <0.001 | 1.53 (1.20, 1.95) | <0.001 |
| Intellectual disability | 2.22 (0.98, 5.03) | 0.055 | 2.03 (1.07, 3.86) | 0.029 |
| Autism | 0.78 (0.42, 1.44) | 0.423 | 0.69 (0.33, 1.44) | 0.329 |
| **Any use of psychotropic medication** | 1.60 (1.50, 1.70) | <0.001 | 1.67 (1.55, 1.79) | <0.001 |
| Sedatives/hypnotics | 1.50 (1.38, 1.63) | <0.001 | 1.59 (1.48, 1.72) | <0.001 |
| Anxiolytics | 1.53 (1.42, 1.65) | <0.001 | 1.57 (1.45, 1.69) | <0.001 |
| Antidepressants | 1.42 (1.32, 1.52) | <0.001 | 1.45 (1.35, 1.55) | <0.001 |
| Antipsychotics | 1.28 (1.05, 1.55) | 0.013 | 1.28 (1.09, 1.49) | 0.002 |

ADHD, attention-deficit hyperactivity disorder; CI, confidence interval; CIN3+, cervical intraepithelial neoplasia grade 3 or worse; HPV, human papillomavirus; HR, hazard ratio.

HR-HPV, high-risk HPV, including 14 types: 16, 18, 31, 33, 35, 39, 45, 51, 52, 56, 58, 59, 66, and 68.

[1]Adjusted for age, country of birth, educational level, HPV vaccination status, and maternal history of CIN3+.

[2]Tobacco- and alcohol-related disorders are excluded.

the highest rate of loss to follow-up, whereas the result was not statistically significant for tobacco-related disorder and intellectual disability.

HRs were adjusted for age, country of birth, educational level, HPV vaccination status, and maternal history of CIN3+.

Stratification analysis by age did not suggest any difference in the association of mental illness with infection rate of high-risk HPV between age groups (S4 Table). Finally, compared to women with neither a diagnosis of mental disorder nor use of psychotropic mediation, use of psychotropic medication without a specialist diagnosis was associated with a 1.43 times increased infection rate of any high-risk HPV (95% CI [1.33 to 1.55]; $p < 0.001$), whereas women with a specialist diagnosis of mental disorder had a 1.54 times increased infection rate of high-risk HPV (95% CI [1.40, 1.69]; $p < 0.001$) (S5 Table). Sensitivity analysis using 12 months as the grace period showed similar results as the main analysis (S6 Table).

## Discussion

In this population-based cohort study, we found an increased rate of incident infection with high-risk HPV, either HPV-16/18 or other high-risk HPV types, among women with a specialist diagnosis of mental disorder or use of psychotropic medications without a specialist diagnosis. A statistically significant association was noted for the majority of psychiatric disorders and psychotropic medications studied, except tobacco-related disorder, psychotic disorder (specialist diagnosis or use of antipsychotics), and neurodevelopmental disorders. Women with mental illness were also more likely to miss the following screening.

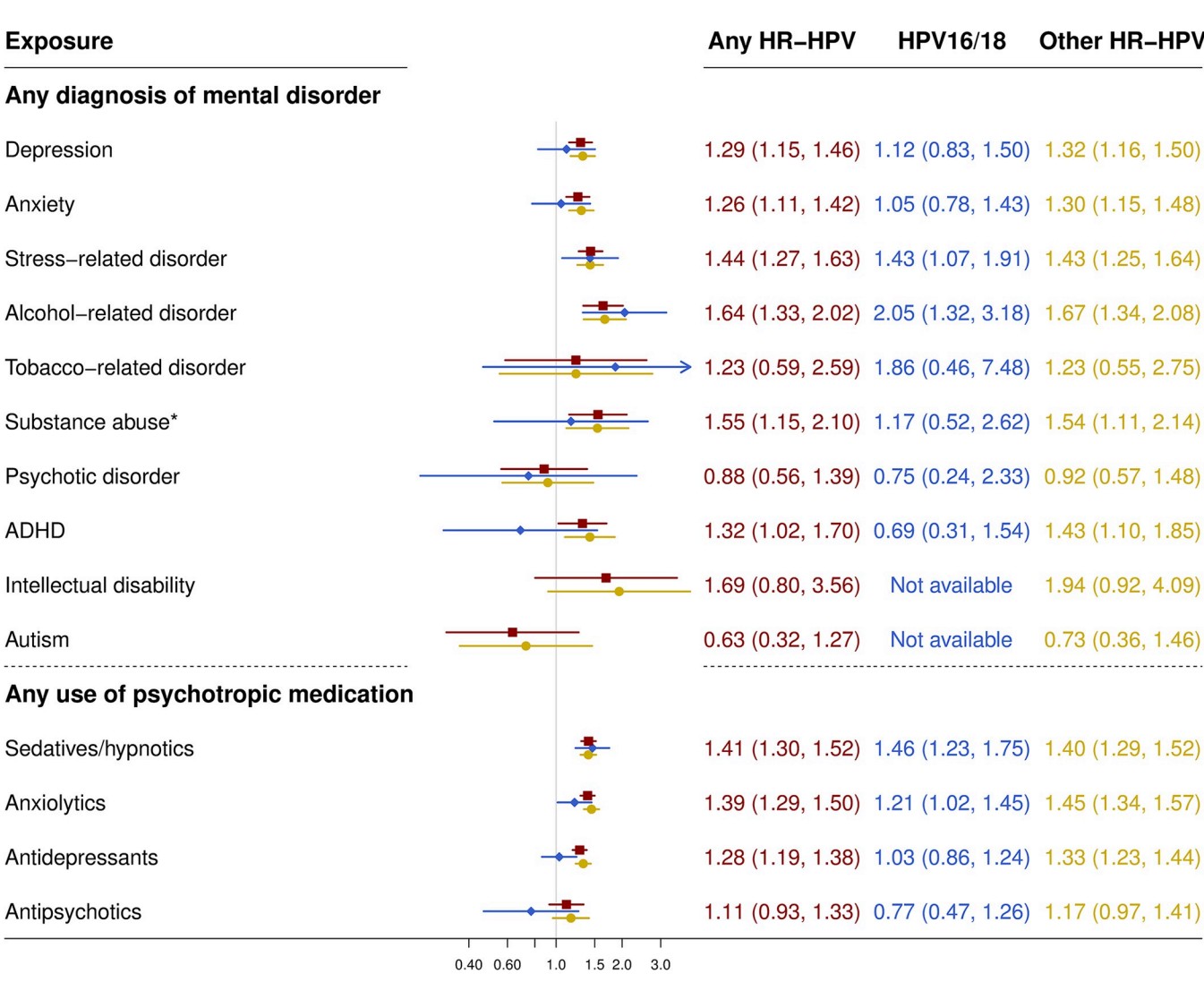

**Fig 2. HRs with 95% CIs of any high-risk HPV infection in relation to any diagnosis of mental disorder or any use of psychotropic medication.** *Tobacco-and alcohol-related disorders were excluded. ADHD, attention-deficit hyperactivity disorder; CI, confidence interval; CIN3+, cervical intraepithelial neoplasia grade 3 or worse; HPV, human papillomavirus; HR, hazard ratio. HR-HPV, high risk-HPV, including 14 types: 16, 18, 31, 33, 35, 39, 45, 51, 52, 56, 58, 59, 66, and 68. HRs were adjusted for age, country of birth, educational level, HPV vaccination status, and maternal history of CIN3+.

The increased infection rate of high-risk HPV was found among women with a specialist diagnosis of mental disorder (i.e., relatively severe mental illness) as well as among women that used psychotropic medications without a specialist diagnosis (i.e., likely relatively milder mental illness). This is in line with our previous study [3] including a population of 4,112,598 women born during 1940 to 1995, who ever resided in Sweden during 1968 to 2018, showing an increased risk of precancerous cervical lesions among both groups of women. The 337,116 women we included in this study were at age 30 to 64, ever lived in Stockholm, and had a negative HPV test, during August 2014 to December 2019. Therefore, the population in this study is a subset of the previous study. The increased infection rate was similar between HPV-16/18 and other high-risk HPV types and was consistent across age groups, indicating a potentially common mechanism linking together mental illness and different high-risk HPV types at

**Table 3. Adjusted HRs with 95% CIs for infection with HPV16/18 or other high risk-HPV in relation to diagnosis of mental disorder or use of psychotropic medication.**

| Groups | Person-years | Any high risk-HPV[1] No. of infection | Adjusted HR (95% CI)[2] | P value | HPV16/18 No. of infection | Adjusted HR (95% CI)[2] | P value | Other high risk-HPV No. of infection | Adjusted HR (95% CI)[2] | P value |
|---|---|---|---|---|---|---|---|---|---|---|
| **Any diagnosis of mental disorder** | | | | | | | | | | |
| No | 668,494 | 2,644 | Reference | | 489 | Reference | | 2,281 | Reference | |
| Yes | 110,648 | 619 | 1.35 (1.23–1.47) | $p < 0.001$ | 108 | 1.27 (1.03–1.57) | 0.025 | 545 | 1.37 (1.25–1.51) | $p < 0.001$ |
| **Any diagnosis of psychiatric disorder** | | | | | | | | | | |
| No | 670,751 | 2,657 | Reference | | 489 | Reference | | 2,294 | Reference | |
| Yes | 108,391 | 606 | 1.35 (1.23–1.47) | $p < 0.001$ | 108 | 1.31 (1.06–1.61) | 0.013 | 532 | 1.37 (1.25–1.51) | $p < 0.001$ |
| **Any diagnosis of neurodevelopmental disorder** | | | | | | | | | | |
| No | 767,752 | 3,193 | Reference | | 591 | Reference | | 2,760 | Reference | |
| Yes | 11,389 | 70 | 1.25 (0.99–1.59) | 0.062 | 6 | 0.58 (0.26–1.30) | 0.184 | 66 | 1.37 (1.07–1.75) | 0.013 |
| **Any use of psychotropic medication** | | | | | | | | | | |
| No | 449,011 | 1,639 | Reference | | 316 | Reference | | 1,403 | Reference | |
| Yes | 330,130 | 1,624 | 1.44 (1.34–1.54) | $p < 0.001$ | 281 | 1.28 (1.09–1.50) | 0.003 | 1,423 | 1.48 (1.37–1.59) | $p < 0.001$ |

CI, confidence interval; CIN3+, cervical intraepithelial neoplasia grade 3 or worse; HPV, human papillomavirus; HR, hazard ratio.

[1]High-risk HPV includes 14 types: 16, 18, 31, 33, 35, 39, 45, 51, 52, 56, 58, 59, 66, and 68.

[2]Adjusted for age, country of birth, educational level, HPV vaccination status, and maternal history of CIN3+.

different ages. Moreover, the increased infection rate of high-risk HPV was independent of HPV vaccination and maternal history of CIN3+, suggesting that mechanisms other than differential vaccination uptake and genetic susceptibility to cervical cancer might contribute. For example, it has been suggested that women with mental illness are more likely to contract sexually transmitted diseases, have sexual debut at an early age [8], and smoke [9]. Several illegal drugs, including cannabis, may exhibit immunosuppressive effects, potentially increasing the susceptibility to viral infections [24]. Notably, a higher proportion of individuals with mental illness use these substances compared to the general population [25]. In addition, women with mental illnesses are more likely to have experienced adverse life events such as childhood sexual abuse [26], a traumatic life event previously shown to be associated with both the development of mental ill health and an elevated risk of HPV infection [11,27]. Although the precise mechanisms remain unknown, immune alterations might have contributed, in acquiring, clearing, and reactivating of HPV infection [26,27]. Although not all factors are modifiable, targeted interventions are possible and feasible in order to minimize such disparities.

The present study showed that, even in a cohort of women who participated in cervical screening and had a negative test of high-risk HPV to start with, women with mental illness were still more likely to miss the following screening and thus less likely to have their HPV infection detected. Therefore, HRs observed in this study might be an underestimate of the real association between mental illness and HPV infection, partly explaining the lack of statistically significant findings in the right-censored analysis for several mental illnesses, including psychotic disorder, substance abuse, and neurodevelopmental disorders. This is further consistent with our previous finding that women with psychotic disorder, substance abuse, and neurodevelopmental disorders have the lowest degree of screening participation [3].

A few studies have examined the risk of HPV infection in relation to mental illness, primarily substance abuse, and reported conflicting results [28–30]. Two studies [28,30] found no association between drug abuse and the risk of HPV infection, whereas 1 study reported an increased risk of infection with high-risk HPV in relation to cocaine use [29]. These studies all

**Table 4. Differential loss to follow-up between women with and without mental illness.**

| Mental illness | Exposure | Person-years of follow-up | Number of loss to follow-up | Incidence rate of loss to follow-up (per 1,000 person-years) | Adjusted HR of loss to follow-up | P value |
|---|---|---|---|---|---|---|
| Any diagnosis of mental disorder | Unexposed | 668,494 | 26,272 | 39.3 | Reference | |
| | Exposed | 110,648 | 5,589 | 50.51 | 1.26 (1.23–1.3) | <0.001 |
| Any diagnosis of psychiatric disorder | Unexposed | 670,751 | 26,381 | 39.33 | Reference | |
| | Exposed | 108,391 | 5,480 | 50.56 | 1.27 (1.23–1.3) | <0.001 |
| Depression | Unexposed | 726,287 | 29,130 | 40.11 | Reference | |
| | Exposed | 52,854 | 2,731 | 51.67 | 1.27 (1.22–1.32) | <0.001 |
| Anxiety | Unexposed | 727,596 | 29,131 | 40.04 | Reference | |
| | Exposed | 51,545 | 2,730 | 52.96 | 1.28 (1.23–1.33) | <0.001 |
| Stress-related disorder | Unexposed | 734,444 | 29,454 | 40.1 | Reference | |
| | Exposed | 44,698 | 2,407 | 53.85 | 1.3 (1.24–1.35) | <0.001 |
| Alcohol-related disorder | Unexposed | 765,934 | 31,168 | 40.69 | Reference | |
| | Exposed | 13,208 | 693 | 52.47 | 1.35 (1.25–1.46) | <0.001 |
| Tobacco-related disorder | Unexposed | 777,770 | 31,793 | 40.88 | Reference | |
| | Exposed | 1,372 | 68 | 49.55 | 1.25 (.98–1.58) | 0.068 |
| Substance abuse* | Unexposed | 772,898 | 31,476 | 40.72 | Reference | |
| | Exposed | 6,244 | 385 | 61.66 | 1.5 (1.36–1.66) | <0.001 |
| Psychotic disorder | Unexposed | 773,505 | 31,590 | 40.84 | Reference | |
| | Exposed | 5,637 | 271 | 48.08 | 1.23 (1.1–1.39) | <0.001 |
| Any diagnosis of neurodevelopmental disorder | Unexposed | 767,752 | 31,172 | 40.6 | Reference | |
| | Exposed | 11,389 | 689 | 60.49 | 1.38 (1.28–1.49) | <0.001 |
| ADHD | Unexposed | 769,645 | 31,282 | 40.64 | Reference | |
| | Exposed | 9,497 | 579 | 60.97 | 1.39 (1.28–1.51) | <0.001 |
| Intellectual disability | Unexposed | 778,199 | 31,809 | 40.88 | Reference | |
| | Exposed | 943 | 52 | 55.15 | 1.28 (.97–1.68) | 0.078 |
| Autism | Unexposed | 776,606 | 31,698 | 40.82 | Reference | |
| | Exposed | 2,536 | 163 | 64.28 | 1.47 (1.26–1.71) | <0.001 |
| Any use of psychotropic medication | Unexposed | 449,011 | 17,598 | 39.19 | Reference | |
| | Exposed | 330,130 | 14,263 | 43.2 | 1.14 (1.11–1.16) | <0.001 |
| Hypnotics and sedatives | Unexposed | 586,437 | 23,565 | 40.18 | Reference | |
| | Exposed | 192,704 | 8,296 | 43.05 | 1.13 (1.11–1.16) | <0.001 |
| Anxiolytics | Unexposed | 583,404 | 23,108 | 39.61 | Reference | |
| | Exposed | 195,738 | 8,753 | 44.72 | 1.15 (1.12–1.17) | <0.001 |
| Antidepressants | Unexposed | 568,818 | 22,239 | 39.1 | Reference | |
| | Exposed | 210,323 | 9,622 | 45.75 | 1.17 (1.15–1.2) | <0.001 |
| Antipsychotics | Unexposed | 751,870 | 30,537 | 40.61 | Reference | |
| | Exposed | 27,272 | 1,324 | 48.55 | 1.22 (1.15–1.29) | <0.001 |

ADHD, attention-deficit hyperactivity disorder; CIN3+, cervical intraepithelial neoplasia grade 3 or worse; HPV, human papillomavirus; HR, hazard ratio.

*Tobacco- and alcohol-related disorders are excluded.

used questionnaire data and had limited sample size. For example, although 1 study [30] did suggest a higher prevalence of HPV infection among women with drug abuse (37.7%) than women without drug abuse (21.9%), no statistically significant association was found after adjusting for lifestyle factors, including age at onset of sexual activity, number of sexual partners, uptaking of cervical screening via Pap smear, and current smoking. It is unclear, however, whether these lifestyle factors should be adjusted for as they are likely mediating the causal pathways between mental illness and HPV infection. In contrast to these studies, we

focused on specialist diagnosis of mental disorder, as a proxy for severe mental illness, as well as use of psychotropic medication without specialist diagnosis, as a proxy for less severe mental illness, and extended the existing evidence base from substance abuse to all major psychiatric disorders as well as neurodevelopmental disorders.

Strengths of our study include the large sample size and comprehensive data on clinical diagnoses of mental disorders, filled prescription of psychotropic medications, HPV screening results, and mother–daughter linkage from the whole female population of the greater Stockholm area. Acquired from national registers, these data are therefore less affected by selection and information bias. Importantly, the 337,116 women included in the present study are a subset of the 4 million women included in our previous study [3] and the findings corroborated with each other. However, there are a few limitations. First, as we could only detect HPV infection through screening, we might have missed infections acquired and cleared between 2 screening rounds, or infections undetected due to unattended screening. Additionally, as the eligible participants of the study must have had a negative test result for HPV infection, the study participants were likely at a lower risk of HPV infection compared to other women. As a result of both, the overall infection rate of high-risk HPV infection appeared lower in the present study, compared to previous studies [31,32]. Given that we have found women with mental illness are less likely to attend for the next screening, they tend to have longer follow-up after their last negative test, during which they are assumed to be infection-free; thus, the infection rate among women with mental illness and the overall association might be underestimated. To address this, we conducted an interval-censored analysis on women with at least 2 HPV tests, restricting the follow-up as between the first and last tests, which showed similar but slightly stronger results. However, the right-censored analysis provided the opportunity to study the entire sample of the present study, instead of only women with at least 2 HPV tests during the study, and the differential adherence to HPV screening in relation to mental illness. Second, women included in the study cohort were only a subset of the female population in Stockholm, as they must have at least 1 negative test result of HPV during the study period. However, we found no substantial difference in the proportion of mental illness between the study cohort and the entire female population in Stockholm. This argues against substantial selection bias and suggests that our results are generalizable. Lastly, we had a relatively short follow-up as HPV-based screening was only introduced in 2014 in Stockholm. As a result, the majority (82%) of women in our study did not have had the time to receive a new screening. However, Stockholm was among the earliest regions to initiate HPV-based screening in Europe [33], and our study represents the largest study to date with the longest follow-up concerning women regularly screened for HPV infection. Regardless, follow-up studies with longer follow-up or larger sample size (e.g., national data) are needed to validate these findings with better statistical power, especially for subgroup analysis.

Based on our previous study of the entire Swedish female population [3], women with mental illness participate less in cervical screening and experience a higher risk of cervical cancer. The fact that there is an increased risk of infection with high-risk HPV in relation to mental illness among women who participate in screening indicates that both factors—lower screening attendance and higher risk of acquiring high-risk HPV infection—contribute to the increased risk of invasive cervical cancer in relation to mental illness. We ended the present study in 2019 as the COVID-19 pandemic impacted greatly the HPV screening program, due to reduced testing capacity and screening participation. Fortunately, self-collected HPV tests have been introduced to enhance cervical screening since July 2020 in Sweden [34].

In conclusion, we observed an increased infection rate of high-risk HPV among women with a specialist diagnosis of psychiatric disorder or neurodevelopmental disorder or with filled prescriptions of psychotropic medications, despite that they were less likely to attend

screening compared with other women. This timely information may guide interventions to achieve the WHO cervical cancer elimination agenda.

## Supporting information

**S1 STROBE checklist. Checklist of items that should be included in reports of observational studies.**
(DOCX)

**S1 Statistical Analysis Plan. Risk of HPV infection in women with mental disorders.**
(DOCX)

**S1 Fig. Unadjusted HRs with 95% CIs of any high-risk HPV infection in relation to diagnosis of mental disorder or use of psychotropic medication.** *Tobacco- and alcohol-related disorders are excluded. ADHD, attention-deficit hyperactivity disorder; CI, confidence interval; HPV, human papillomavirus; HR, hazard ratio. HR-HPV, high-risk HPV, including 14 types: 16, 18, 31, 33, 35, 39, 45, 51, 52, 56, 58, 59, 66, and 68.
(PDF)

**S1 Table. International Classification of Diseases (ICD) codes for predefined mental disorders.**
(DOCX)

**S2 Table. ATC codes for predefined psychotropic medications.**
(DOCX)

**S3 Table. Number of person-years by different characteristics among women with or without diagnosis of mental disorder or filled prescription of psychotropic medication.**
(DOCX)

**S4 Table. Adjusted HRs with 95% CIs of high-risk HPV infection in relation to diagnosis of mental disorder or use of psychotropic medication, by attained age.** CI, confidence interval; HPV, human papillomavirus; HR, hazard ratio.
(DOCX)

**S5 Table. IRs and adjusted HRs with 95% CIs of high-risk HPV infection by severity of mental illness.** CI, confidence interval; HPV, human papillomavirus; HR, hazard ratio; IR, incidence rate.
(DOCX)

**S6 Table. Adjusted HRs with 95% CIs of high-risk HPV infection in relation to diagnosis of mental disorder or use of psychotropic medication, using 12-month grace period to define loss to follow-up.** CI, confidence interval; HPV, human papillomavirus; HR, hazard ratio.
(DOCX)

## Author Contributions

**Conceptualization:** Unnur Valdimarsdóttir, Fang Fang.

**Data curation:** Jiangrong Wang, Karin Sundström.

**Formal analysis:** Eva Herweijer.

**Funding acquisition:** Fang Fang.

**Investigation:** Eva Herweijer, Jiangrong Wang, Donghao Lu.

**Methodology:** Eva Herweijer, Kejia Hu, Jiangrong Wang.

**Project administration:** Karin Sundström, Fang Fang.

**Resources:** Jiangrong Wang, Pär Sparén, Karin Sundström.

**Software:** Eva Herweijer, Kejia Hu.

**Supervision:** Fang Fang.

**Validation:** Kejia Hu.

**Visualization:** Kejia Hu.

**Writing – original draft:** Eva Herweijer, Kejia Hu.

**Writing – review & editing:** Eva Herweijer, Kejia Hu, Jiangrong Wang, Donghao Lu, Pär Sparén, Hans-Olov Adami, Unnur Valdimarsdóttir, Karin Sundström, Fang Fang.

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
