## [Editor Report · Decision Letter 0]

21 Aug 2023

Dear Dr Hu, 

Thank you for submitting your manuscript entitled "Incidence of oncogenic HPV infection in women with and without mental illness: a population-based cohort study in Sweden" for consideration by PLOS Medicine.

Your manuscript has now been evaluated by the PLOS Medicine editorial staff as well as by an academic editor with relevant expertise and I am writing to let you know that we would like to send your submission out for external peer review.

Please re-submit your manuscript within two working days, i.e. by Aug 24 2023 11:59PM.

Kind regards,

Katrien Janin, PhD

Senior Editor

PLOS Medicine

---

## [Decision Letter · Decision Letter 1]

13 Oct 2023

Dear Dr. Hu,

Thank you very much for submitting your manuscript "Incidence of oncogenic HPV infection in women with and without mental illness: a population-based cohort study in Sweden" (PMEDICINE-D-23-02359R1) for consideration at PLOS Medicine. 

[LINK]

In light of these reviews, I am afraid that we will not be able to accept the manuscript for publication in the journal in its current form, but we would like to consider a revised version that addresses the reviewers' and editors' comments. Obviously we cannot make any decision about publication until we have seen the revised manuscript and your response, and we plan to seek re-review by one or more of the reviewers. 

We expect to receive your revised manuscript by Nov 03 2023 11:59PM. Please email us (plosmedicine@plos.org) if you have any questions or concerns.

We look forward to receiving your revised manuscript. 

Sincerely,

Katrien Janin, PhD

PLOS Medicine

plosmedicine.org

Comments from the Executive editor: Please outline in detail whether and how much this population overlaps with the population studied in the prior Lancet Pub Health paper. It follows that these patients might have been a sub-set of those in the prior paper, and this will be important to acknowledge/discuss. The points made by the statistical reviewer are also very important.

GENERAL: 

Please provide 95% CIs and p values for all results were appropriate (including the abstract), check and amend throughout. We suggest reporting statistical information in the following format: ‘x’; (95% CI [‘y’,’ z’] p value) For p values, please report as p<0.001 and where higher as 'p=0.002'. Please add the statistical method used to your method section.

For in-text reference, citations are placed within square parentheses and should precede punctuation. Please amend throughout.

STUDY DESIGN:

Please ensure that the study is reported according to the STROBE guideline, and include the completed STROBE checklist as Supporting Information. Please add the following statement, or similar, to the Methods: ""This study is reported as per the Strengthening the Reporting of Observational Studies in Epidemiology (STROBE) guideline (S1 Checklist).""

When completing the checklist, please use section and paragraph numbers, rather than page numbers."

"Did your study have a prospective protocol or analysis plan? Please state this (either way) early in the Methods section.

c) In either case, changes in the analysis-- including those made in response to peer review comments-- should be identified as such in the Methods section of the paper, with rationale."

For all observational studies, in the manuscript text, please indicate: (1) the specific hypotheses you intended to test, (2) the analytical methods by which you planned to test them, (3) the analyses you actually performed, and (4) when reported analyses differ from those that were planned, transparent explanations for differences that affect the reliability of the study's results. If a reported analysis was performed based on an interesting but unanticipated pattern in the data, please be clear that the analysis was data-driven.

DATA AVAILABILITY:

The Data Availability Statement (DAS) requires revision. Please include an appropriate contact as a study author cannot be the contact person for the data.

ABSTRACT:

Abstract Methods and Findings:

Please include length of follow up

Please quantify the main results (with 95% CIs and p values - in line with the comment made above).

Please clarify when you write ‘risk’ if you are referring to absolute or relative risk

Result: given the study design, it would be beneficial to use associational language for the outcome.

AUTHORS SUMMARY:

Ideally each sub-heading should contain 2-3 single sentence, concise bullet points containing the most salient points from your study.

In the final bullet point of ‘What Do These Findings Mean?’ Please include the main limitations of the study in non-technical language.

METHODS:

Please add the statistical method used to your method section.

Please add the following statement, or similar, to the Methods: ""This study is reported as per the Strengthening the Reporting of Observational Studies in Epidemiology (STROBE) guideline (S1 Checklist).

RESULTS: 

As above, please provide 95% CIs and p values for all results were appropriate 

ACKNOWLEDGMENTS/ DECLARATIONS

Please remove all statements apart from acknowledgements, author contributions and abbreviations from the end of the main manuscript and include these only in the relevant parts of the manuscript submission form. Funding, competing interest, and data availability will be compiled as metadata.

REFERENCES:

Please use the "Vancouver" style for reference formatting, and see our website for other reference guidelines https://journals.plos.org/plosmedicine/s/submission-guidelines#loc-references

Please ensure that in the bibliography up to but no more than 6 author names are listed, followed by et al., in the event that more than 6 authors contribute to an individual study. Journal name abbreviations should be those found in the National Center for Biotechnology Information (NCBI) databases.

Please also ensure that any references to online-only sources include a date of accession, and in the reference list, please convert all italics to plain text.

FIGURES: 

Figure 2: Please consider avoiding the use of red and green in order to make your figure more accessible to those with colour blindness. Likewise for figure S1. 

Comments from the reviewers:

Reviewer #1: Thank you for the opportunity to review this timely and important paper on an under-researched area examining the relationship between HPV infection acquisition by women with and without mental illness within the context of an HPV based screening program. This question is pertinent in the context of the global quest for the elimination of cervical cancer.

The paper is well written and concise and fills important gaps in the literature. The figures and tables are well portrayed.

The paper's strengths lie in its robust approach through analysis of participants in randomised implementation trials and Sweden's data linkage capability to link HPV screening results to mental illness diagnoses/use of psychotropic medications, and to HPV vaccination and maternal history of cervical disease.

However, the limitations require further consideration, in particular the choice of a relatively short 6-month period of follow up which inevitably means that a sizeable group of participants with mental illness are deemed 'lost to follow up' which will likely lead to potential underestimation of HPV infection (ie the primary outcome). The relatively sample size relative to the Stockholm population is well made due to the requirement for a negative HPV results to enter the trial and the relatively short follow up period due to the relative recency of the introduction of HPV screening in Sweden. 

The approach to defining who has and hasn't got mental illness has been well described in a previous paper.

The approach to inclusion is well described and the end points of participation are clearly defined.

I have some comments and some suggestions for improvement - I also note that I am not an expert statistician/epidemiologist so strongly recommend expert review in these areas given the relative complexity of their methodological approaches.

A couple of general comments:

1. While not a criticism of the paper the rate of new infections (incidence rate) of 1% seems relatively low?

2. The data outcomes are generally robust and the sample from the randomised trial are generally representative of the national population, however, it could be argued that it could have been preferable to wait until more national HPV data becomes available for analysis to increase the size of some of the groups?

Specific comments:

3. The information on page 4/ 5 regarding the randomized implementation trials would benefit from a little more context - e.g. were these whole of Stockholm population trials? 

4. A brief explanation about how the screening program is organised and in particular how/when reminders/recalls are sent would be useful (especially considering the point below re the challenges of the very short 6 month time frame for re-screening after which the person is deemed lost to follow up and excluded from the study)?

5. The chosen period of 6 months grace for women to return for their follow up 3 or 5-yearly HPV screening test before being deemed lost to follow up is very short (as noted by the authors) and needs explanation as to why it was chosen rather than, say, 12 months?

 This is highly relevant (also noted by the authors) for those groups with certain mental illness diagnoses (particularly those with neurodevelopmental disorders including autism) where multiple barriers may preclude such timely access. While the authors have noted this may lead to an underestimation of HPV incident infection in these groups, this seems to be a major challenge for the central premise of the paper given that these groups may also be more likely to have experienced sexual abuse than the general population which may go un-noted without sufficient time to present for follow up screening since their previous negative test? As also noted this may lead to the lack of statistical power and statistically significant findings for several mental illnesses.

In short, the implications for people with a mental illness diagnosis being less likely to be screened but more likely to acquire HPV would benefit from greater scrutiny in the paper (noting that a focus on timely screening may be most relevant from a policy perspective?)

6. Another argument that could be considered relates to latency of HPV acquired at an earlier age (eg due to child sexual abuse), whereby the HPV is undetectable at the time of screening (and in this case at the time of the prior screen) and then reactivated later with a decline in immune function and goes on to cause disease - this may be pertinent given the relatively high rates of sexual abuse for some groups under the mental illness umbrella? 

7. The argument around biological plausibility for increased HPV acquisition due to immune suppression and other mechanisms is well made - ie it is not all behavioural/modifiable?

Minor points

8. Page 4 line 38 - the term vulnerable is generally not preferred as it implies weaknesses on the part of the individual rather than society - it could be replaced by a term such as marginalised/under-served?

9. There is a minor typo dirode

Reviewer #2: Herweijer et al. have conducted a population-based cohort study in Sweden to investigate the incidence of oncogenic HPV infection in women with and without mental illness. The study aimed to determine whether women with mental illness have a higher risk of HPV infection compared to those without mental illness.

Using a cohort of 337,116 women aged 30-64 living in Stockholm, who had previously tested negative for 14 high-risk HPV subtypes in HPV-based screening from August 2014 to December 2019, the researchers identified women with mental illness based on specialist diagnoses of mental disorders or filled prescriptions for psychotropic medication.

The results revealed that women with mental illness had an increased risk of HPV infection compared to women without mental illness. Specifically, those with a specialist diagnosis of mental disorder had a hazard ratio (HR) of 1.35 (95% confidence interval [CI]: 1.23-1.47), while those with filled prescriptions for psychotropic medication had an HR of 1.44 (95% CI: 1.34-1.54). This elevated risk was observed across various mental health conditions, including depression, anxiety, stress-related disorders, and substance-related disorders, as well as among users of antidepressants, anxiolytics, sedatives, and hypnotics. Importantly, these findings held true across different age groups.

However, the study had certain limitations, such as the selection of the female population in Stockholm and the relatively short follow-up period due to the introduction of HPV-based screening in 2014.

In conclusion, this study suggests that women with mental illness are at a higher risk of HPV infection. These findings underscore the need for targeted strategies to address cervical cancer risk in this vulnerable population, aligning with the WHO's agenda for the global elimination of cervical cancer.

The claims are properly placed in the context of the previous literature. The experimental data support the claims. The manuscript is written clearly enough that most of it is understandable to non-specialists. The authors have provided adequate proof for their claims, without overselling them. The authors have treated the previous literature fairly. The paper offers enough details of methodology so that the experiments could be reproduced.

Minor revisions

Page 4, line 43-44, "It is therefore likely that mental illness influences the risk of cervical cancer development also through other mechanisms like HPV infection."

Unclear. Please reformulate. It is important to note that HPV is the primary cause of cervical cancer in both women with mental illnesses and all other women.

Page 4, line 46-69, add "sexual abuse"

"An elevated risk of acquiring HPV infection is indeed plausible among women with mental illness, due to biological and behavioral factors. For instance, women with mental illness have been suggested to demonstrate abnormal levels of immune biomarkers, engage in riskier sexual behaviors, be more likely to smoke, have limited knowledge of HPV infection, and may also have a higher prevalence of sexual abuse compared to other women."

Alternative introduction:

"Cervical cancer, primarily caused by oncogenic human papillomavirus (HPV) infections on the cervix, is preventable through HPV vaccination and cervical screening, facilitating early detection and treatment of precancerous lesions. In 2020, the World Health Organization (WHO) launched efforts to eliminate cervical cancer as a public health issue. However, women with mental illness present a unique challenge to this agenda, as they not only face an increased risk of cervical cancer but also tend to have reduced participation in cervical screening.

This heightened risk among women with mental illness cannot be solely attributed to lower screening rates. Even among those who participate in screening, women with specialist diagnoses of mental disorders still carry a twofold risk of developing cervical precancerous lesions. This suggests that mental illness likely influences cervical cancer risk through other mechanisms, including the potential impact of HPV infection.

Several factors contribute to the plausible elevated risk of HPV infection among women with mental illness, encompassing abnormal levels of immune biomarkers, engagement in riskier sexual behaviors, higher smoking rates, limited knowledge about HPV infection, and a potentially higher incidence of sexual abuse compared to their counterparts. Despite these concerns, population-based studies examining HPV infection disparities between women with and without mental illness remain limited.

To address this gap, we conducted a study using data from a substantial prospective cohort of women participating in HPV-based cervical screening in Stockholm, Sweden. Our hypothesis was that women with mental illness would exhibit an increased risk of HPV infection."

Page 5, line 64-65, "The trial initially focused on women aged 56-60 but later expanded into a comprehensive randomized implementation that encompassed women aged 30-64 starting from August 2014."

Page 5, line 67-68, "As a result, beginning in 2017, all women between the ages of 30 and 64 in the greater Stockholm area underwent primary screening using HPV-based testing."

Page 5, line 68-69, "The screening test covers 14 high-risk HPV types, including HPV-16, 18, 31, 33, 35, 39, 45, 51, 52, 56, 58, 59, 66, and 68."

Page 7, line 140, "We employed attained age as the fundamental time scale for our analysis."

Page 10, line 227, add "A significant number of women with mental illnesses have experienced childhood sexual abuse. This traumatic experience can lead to the development of mental health disorders and is also associated with an elevated risk of both HPV infection and cervical cancer."

Bulik, C., Prescott, C., & Kendler, K. (2001). Features of childhood sexual abuse and the development of psychiatric and substance use disorders. The British Journal of Psychiatry, 179(5), 444-449. doi:10.1192/bjp.179.5.444

Cao CD, Merjanian L, Pierre J, Balica A. A Discussion of High-Risk HPV in a 6-Year-Old Female Survivor of Child Sexual Abuse. Case Rep Obstet Gynecol. 2017;2017:6014026. doi: 10.1155/2017/6014026. Epub 2017 May 23. PMID: 28620555; PMCID: PMC5460386.

Page 10, line 227, "Several illegal drugs, including cannabis, may exhibit immunosuppressive effects, potentially increasing the susceptibility to viral infections. Notably, a higher proportion of individuals with mental illness use these substances compared to the general population."

Page 12, line 276, "Fortunately, self-collected HPV tests have been introduced to enhance cervical screening since July 2020 in Sweden."

Page 20-22, In Table 2 and Table 3, for enhanced readability, please use thousand separators in large numbers (e.g., 668494 => 668,494).

Reviewer #3: Alex McConnachie, Statistical Review

This review looks at the statistical elements of the paper by Herweijer et al, which presents data from linked routine datasets to assess the association between mental illness (hospital diagnosis, or determined by prescription history), and high-risk HPV infection, detected through screening in Stockholm, Sweden.

On the face of it, the methods look very good, with Cox models used to assess associations whilst adjusting for potential confounders. The PH assumption is checked; various subgroup and sensitivity analyses are reported, and the results are presented quite clearly, and interpreted appropriately.

However, there are one or two issues with the analysis that I can see.

The study cohort is defined as those women living in the study area between 01/08/2014 and 31/12/2019, aged 30-64, with a negative test for HPV during this period. Date of entry to the cohort is defined as the latest of their first negative test, their 30th birthday, or 01/08/2014. Given the cohort definition, is this essentially the same as saying that women enter the cohort at their first negative test?

Since someone must have a negative test during the time window, their first negative test must come after 01/08/2014, so it seems redundant to include 01/08/2014 in the definition. For anyone who turns 30 during the study period, their HPV status at that point is unknown, regardless of whether they had a previous test result, so it would make more sense to make the start date equal to the first negative test during the study period after their 30th birthday.

There is a similar problem in defining the date of censoring. This is defined as the earliest of their 65th birthday, emigration (from Stockholm, or from Sweden), death, 31/12/2019, or one screening interval plus 6 months after their last HPV test (defined as lost to follow-up). This is a major issue; for all those people who are censored, the Cox model will assume that the event (HPV infection) has not occurred, when in fact, the infection status of people who are censored is unknown. They may or may not have been infected since their last negative test result.

The authors look at the probability of loss to follow-up and show that those with prior mental illness are more likely to exit the study in this way, and suggest this may lead to underestimation of the association of interest. This is true, but the same issue will apply to all the other forms of censoring. In the model, anyone who is censored is contributing time at risk, during which it is assumed that there was no infection. In reality, the last date at which it is known that there has been no HPV infection is the date of the last negative test.

Also, the current analysis treats the date of the first positive test as the event date. The analysis is therefore estimating the association between mental illness and the detection of HPV infection, as part of the screening programme. In reality, a positive test indicates that infection occurred at some point since the last negative test. Therefore, the data are interval censored: a negative test indicates that no infection occurred since the previous negative test (though this is probably an assumption, if people can become infected and then clear the infection); a positive test indicates that an infection has occurred since the last negative test.

Therefore, I suggest that the analysis should be redone, defining the date of entry to the cohort as the date of the first negative test, and the date of censoring as the date of the last negative test, and treating the data as interval censored. I believe this will affect the time at risk, but should have no impact on the number of events, so it may have limited impact on the power of the analysis. It may not even affect the overall conclusions very much, but it could.

Other comments

If age is the time scale for analysis, why do the models adjust for age?

In Table 1, is the data for the Greater Stockholm region actually the whole population, or the whole population minus the study cohort? Would it better for the two columns to show distinct populations?

In Supplemental Table 4, data is reportedly shown by "age at infection". Is this correct, assuming not everyone has an infection?

[LINK]

---

## [Decision Letter · Decision Letter 2]

16 Dec 2023

Dear Dr. Hu,

Thank you very much for submitting your manuscript "Incidence of oncogenic HPV infection in women with and without mental illness: a population-based cohort study in Sweden" (PMEDICINE-D-23-02359R2) for consideration at PLOS Medicine. 

[LINK]

In light of these reviews, I am afraid that we will not be able to accept the manuscript for publication in the journal in its current form, but certainly would like to consider a revised version that addresses the reviewers' and editors' comments.

We expect to receive your revised manuscript by Dec 22 2023 11:59PM. Please email us (plosmedicine@plos.org) if you have any questions or concerns.

We ask every co-author listed on the manuscript to fill in a contributing author statement, making sure to declare all competing interests. If any of the co-authors have not filled in the statement, we will remind them to do so when the paper is revised. If all statements are not completed in a timely fashion this could hold up the re-review process. If new competing interests are declared later in the revision process, this may also hold up the submission. Should there be a problem getting one of your co-authors to fill in a statement we will be in contact. You can see our competing interests policy here: http://journals.plos.org/plosmedicine/s/competing-interests.

We look forward to receiving your revised manuscript. 

Sincerely,

Katrien Janin, PhD

PLOS Medicine

plosmedicine.org

The editorial team’s view aligns with those of the statistical reviewer. Please carefully revise your manuscript and ensure you implement the comments by the statistical reviewer in full. 

For your data availability statement, you write: “The data underlying the results presented in the study are available from Statistics Sweden (www.scb.se/en) and The National Board of Health and Welfare (www.socialstyrelsen.se/en). Updated contact information to access the data can be found through the website.” - please add the contact details to your statement.

You also write: “ The raw datasets are however unavailable for sharing because of privacy policies and regulations in Sweden.” We appreciated there may be legal restriction the prevent you from sharing raw identifiable data. Is it possible to share the de-identified data? And if not - please also add why not.

We find the explanation regarding the Lancet paper satisfactory, although we think this needs expanding in the manuscript itself to be completely transparent regarding data overlap

As a last point, to help us extend the reach of your research, please provide any X (formerly known as Twitter) handle(s) that would be appropriate to tag, including your own, your co-authors’, your institution, funder, or lab. Please enter in the submission form any handles you wish to be included when we post about this paper.

Feel free to contact me directly at kjanin@plos.org if you have any questions about the above,

Comments from the reviewers:

Reviewer #2: All comments have been addressed. The response to the comments is adequate and improved the manuscript.

Reviewer #3: Alex McConnachie, Statistical Review

I thank the authors for their responses to my previous comments.

Some sensitivity analyses have been added. The interval censored analysis shows stronger associations, but the description in the text was slightly confusing. To say that infection could happen "…any time after the last negative test" is strange, since the time after the last negative test should not be used in an interval censored analysis. In fact, I would argue that this observation period should not be used in any analysis of infections, because the infection status at the end of this period is unknown.

This also highlights the issue of the period before the first negative test. I do not believe that this should be considered as part of the study. I would hope that this period is excluded from the interval censored analysis, at least. Can the authors confirm that the interval censored analysis only uses periods of observation that start and end with an HPV test?

This raises the question of how many tests people had during the study period. I could be wrong, but I do not recall seeing this information presented in the paper. Logically, only those women with at least two tests (the first being negative) should be included in analyses of HPV infection incidence.

Another analysis has been added defining loss to follow-up as one screening period plus 12-months, rather than 6 months, but I feel this is missing the point. The issue is that after the last HPV test, there is no information about HPV status. By censoring at some point after the last negative test (wherever that is), it is assumed that the individual remains HPV-negative during that time. As far as I can see, the only sensible censoring point is the date of the last negative test.

Saying that, the period after the last negative test is of interest for the analysis of loss to follow-up.

[LINK]

---

## [Decision Letter · Decision Letter 3]

21 Feb 2024

Dear Dr. Hu,

Thank you very much for re-submitting your manuscript "Incidence of oncogenic HPV infection in women with and without mental illness: a population-based cohort study in Sweden" (PMEDICINE-D-23-02359R3) for review by PLOS Medicine.

I have discussed the paper with my colleagues and the academic editor and it was also seen again by the statistical reviewer. I am pleased to say that provided the remaining editorial and production issues are dealt with we are planning to accept the paper for publication in the journal.

[LINK]

We expect to receive your revised manuscript within 10 days. Please email us (plosmedicine@plos.org) if you have any questions or concerns.

We look forward to receiving the revised manuscript by Mar 06 2024 11:59PM.   

Sincerely,

Katrien Janin, PhD

Senior Editor 

PLOS Medicine

plosmedicine.org

Reviewer #3: Alex McConnachie, Statistical Review

I thank the authors once again, for their patience in responding to my comments. I am happy with the answers given and the changes made to the paper.

That said, the authors did not directly respond to my comment about not being able to find the distribution of the number of HPV tests per woman. I really think this should be made much clearer.

Apologies for not spotting this sooner, but on the latest reading, I figured out that the following is available in the flow diagram:

Study population (had at least one negative test): 337116

Had a subsequent positive test: 3263

Had at least one subsequent negative test: 57109

Had no subsequent tests: 276744

In other words, 82% of the study population had a single negative test, any of whom could have acquired HPV after this point, but are assumed not to in the main analyses. It is notable that the interval censored results give confidence intervals of about the same width, despite the smaller sample size, since the number of events is the same. The weaker associations in the main analysis make sense, if women with a history of mental illness are less likely to attend for screening, and so tend to have longer follow-up after their last (or only) negative test, during which they are assumed to be event-free, thus diluting the overall association.

Personally, I feel that the interval censored analysis, including 57109+3263=60372 women, is a more robust approach, but at least this is given in the supplements, and the overall conclusions of the main analysis appear correct, if a little biased.

Editorial comments.

We thank the authors for their extensive and comprehensive revision. 

1. We agree with the view of the statistical reviewer (see above), and also find that interval censored analysis, including 57109+3263=60372 women, is a more robust approach. As such, we like to suggest that these results are moved the to main body of your manuscript and you revise your manuscript accordingly.

2. For clarity, it may also be worth to include the HPV tests per woman in-text (with call-out to the flow diagram)

Please do not hesitate to contact me directly on kjanin@plos.org if you have any questions about the above or like further clarification. 

best wishes,

Katrien

[LINK]

---

## [Editor Report · Decision Letter 4]

4 Mar 2024

Dear Dr Hu, 

On behalf of my colleagues, I am pleased to inform you that we have agreed to publish your manuscript "Incidence of oncogenic HPV infection in women with and without mental illness: a population-based cohort study in Sweden" (PMEDICINE-D-23-02359R4) in PLOS Medicine.

PRESS

Sincerely, 

Katrien G. Janin, PhD 

Senior Editor 

PLOS Medicine